# Towards A Holistic Landscape of
# Situated Theory of Mind in Large Language Models

**Ziqiao Ma**     **Jacob Sansom**     **Run Peng**     **Joyce Chai**
Computer Science and Engineering Division, University of Michigan
{marstin,jhsansom,roihn,chaijy}@umich.edu

## Abstract

Large Language Models (LLMs) have gener-
ated considerable interest and debate regarding
their potential emergence of Theory of Mind
(ToM). Several recent inquiries reveal a lack of
robust ToM in these models and pose a pressing
demand to develop new benchmarks, as current
ones primarily focus on different aspects of
ToM and are prone to shortcuts and data leak-
age. In this position paper, we seek to answer
two road-blocking questions: (1) How can we
taxonomize a holistic landscape of machine
ToM? (2) What is a more effective evaluation
protocol for machine ToM? Following psycho-
logical studies, we taxonomize machine ToM
into 7 mental state categories and delineate ex-
isting benchmarks to identify under-explored
aspects of ToM. We argue for a holistic and situ-
ated evaluation of ToM to break ToM into indi-
vidual components and treat LLMs as an agent
who is physically situated in environments and
socially situated in interactions with humans.
Such situated evaluation provides a more com-
prehensive assessment of mental states and po-
tentially mitigates the risk of shortcuts and data
leakage. We further present a pilot study in
a grid world setup as a proof of concept. We
hope this position paper can facilitate future
research to integrate ToM with LLMs and of-
fer an intuitive means for researchers to better
position their work in the landscape of ToM.

## 1   Introduction

The term *theory of mind* (ToM, sometimes also
referred to as *mentalization* or *mindreading*) was
first introduced by Premack and Woodruff (1978)
as agents' ability to impute *mental states* to them-
selves and others. Many aspects of human cog-
nition and social reasoning rely on ToM model-
ing of others' mental states (Gopnik and Wellman,
1992; Baron-Cohen, 1997; Gunning, 2018). This
is crucial for understanding and predicting others'
actions (Dennett, 1988), planning over others' be-
liefs and next actions (Ho et al., 2022), and various

forms of reasoning and decision-making (Pereira
et al., 2016; Rusch et al., 2020). Inspired by hu-
man ToM, AI researchers have made explicit and
implicit efforts to develop a machine ToM for *so-
cial intelligence*: AI agents that engage in social
interactions with humans (Krämer et al., 2012;
Kennington, 2022) and other agents (Albrecht and
Stone, 2018). Likewise, a machine ToM can en-
hance agents' capacity for interactions (Wang et al.,
2021), explainable decision-making (Akula et al.,
2022), dialogue communication (Qiu et al., 2022;
Takmaz et al., 2023), and collaborative task plan-
ning (Bara et al., 2023).

Machine ToM has received an increasing amount
of attention in the research community, especially
as the field is reshaped by *large language mod-
els* (LLMs) such as ChatGPT (OpenAI, 2022) and
GPT-4 (OpenAI, 2023). This highlights an ongo-
ing debate and discussion on whether a machine
ToM has emerged in LLMs. While LLMs have
demonstrated some capability of inferring commu-
nicative intentions, beliefs, and desires (Andreas,
2022; Kosinski, 2023; Bubeck et al., 2023), re-
searchers also reported concerns regarding a lack
of robust *agency* in LLMs for complex social and
belief reasoning tasks (Sap et al., 2022; Shapira
et al., 2023a) and in-context pragmatic communi-
cation (Ruis et al., 2022). Emerged or not emerged,
that remains a question (or may not even be the
central question to ask). In our view, existing eval-
uation protocols do not fully resolve this debate.
Most current benchmarks focus only on a (few)
aspect(s) of ToM, in the form of written stories,
and are prone to data contamination, shortcuts, and
spurious correlations (Trott et al., 2022; Aru et al.,
2023; Shapira et al., 2023a). Prior to embarking on
extensive data collection for new ToM benchmarks,
it is crucial to address two key questions: (1) How
can we taxonomize a holistic landscape of machine
ToM? (2) What is a more effective evaluation pro-
tocol for machine ToM?

To embrace the transformation brought by LLMs and explore their full potential in understanding and modeling ToM, this position paper calls for a holistic investigation that taxonomizes ToM using the *Abilities in Theory of Mind Space* (ATOMS) framework (Beaudoin et al., 2020). After a review of existing benchmarks under this framework, we put forward a situated evaluation of ToM, one that treats LLMs as agents who are physically situated in environments and socially situated in interactions with humans. We hope this paper will offer an intuitive means to identify research priorities and to help gain a deeper understanding of, as well as to effectively utilize, LLMs in ToM modeling for AI agents in the future.

## 2 Large Language Models as Theory of Mind Agents

Since the advent of pre-trained language models, the research community has questioned whether they possess intrinsic mental states to represent the environment (Li et al., 2021; Storks et al., 2021; Hase et al., 2023) and comprehend the mental states of others (Sap et al., 2019; Zhang et al., 2021) through the textual description (observation) of behavioral cues. The relatively recent breakthroughs of LLMs have created many discussions and debates, primarily concerning the extent to which LLMs possess various capabilities required for a machine ToM. In this section, we first survey recent research presenting evidence and counter-evidence for the emergence of ToM in LLMs. We conclude the discussion with the limitations of current evaluation protocols.

### 2.1 Do Machine ToM Emerge in LLMs?

**Evidence for emergent ToM in LLMs.** Prior to the rise of large language models, there has been growing evidence and acknowledgment of a narrow and limited sense of agency in smaller language models. Andreas (2022) argues that language models have the capacity to predict relations between agents' observations, mental states, actions, and utterances, as they infer approximate representations of beliefs, desires, and intentions of agents mentioned in the context. These representations have a causal influence on the generated text, similar to an intentional agent's state influencing its communicative actions under a Belief-Desire-Intention (BDI) agent model (Bratman, 1987). Amidst the excitement surrounding the release of GPT-4 (Ope-

nAI, 2023), researchers have searched for evidence of an emergent ToM in LLMs. Kosinski (2023) presents 20 case studies each of the unexpected contents task (Perner et al., 1987) and the unexpected transfer (Sally-Anne) task (Baron-Cohen et al., 1985). With direct comparisons to children's performance, the findings have been cited as potential evidence for a spontaneous emergence of ToM in LLMs. Bubeck et al. (2023) present a similar behavioral study with 10 cases of belief, emotion, and intention understanding, concluding that GPT-4 has an advanced level of ToM after qualitative comparison with predecessors. Other case studies have also shown aspects of machine ToM (Li et al., 2023; Holterman and van Deemter, 2023).

**Limitations of ToM capabilities in LLMs.** The above findings contradict the conclusions drawn in Sap et al. (2022)'s earlier study, which shows a clear lack of ToM in GPT-3 (Brown et al., 2020) on SOCIALIQA (Sap et al., 2019) and TOMI (Le et al., 2019) benchmarks. As a potential account, there has been criticism that the cognitive inquiries are anecdotal and inadequate for evaluating ToM in LLMs (Marcus and Davis, 2023; Mitchell and Krakauer, 2023; Shapira et al., 2023a). Following the same evaluation protocol, Ullman (2023) demonstrates that simple adversarial alternatives to Kosinski (2023) can fail LLMs. To further understand if the most recent variants of LLMs possess a robust ToM, Shapira et al. (2023a) present a comprehensive evaluation over 6 tasks and 3 probing methods, showing that a robust machine ToM is absent even in GPT-4 and that LLMs are prone to shortcuts and spurious correlations. Based on the ongoing debate, it can be concluded that, while LLMs exhibit some level of sensitivity at understanding others' mental states, this capability is limited and falls short of achieving robust human-level ToM (Trott et al., 2022; Shapira et al., 2023a).

### 2.2 Roadblocks in ToM Evaluation in LLMs

Given the pressing need for a robust machine ToM in LLMs and large-scale ToM benchmarks, researchers echo several difficulties in the evaluation protocol. Presently, ToM benchmarks suffer from three primary issues summarized as follows.

**Limited aspects of ToM.** The evaluation of machine ToM lacks consistency in the literature due to the ambiguity surrounding the specific mental states being targeted. Existing benchmarks often focus on limited numbers of mental states, such as the *intention* (Yoshida et al., 2008), *belief* (Grant

et al., 2017), *emotion* (Sap et al., 2019), and *knowledge* (Bara et al., 2021) of another agent. While all of these are necessary building blocks of machine ToM, we echo Shapira et al. (2023a)'s concern that the ToM capability of LLMs may have been over-claimed based on evaluations from only a specific aspect of ToM. To give a comprehensive assessment of a holistic machine ToM, a taxonomy is essential to enable researchers to effectively position their work with different focuses and priorities, which may be orthogonal to each other.

**Data contamination.** Data contamination refers to the lack of a verifiable train-test split that is typically established to test the ability of machine learning models to generalize (Magar and Schwartz, 2022). LLMs typically learn from internet-scale data, potentially giving them access during training to the data used to test them (Bubeck et al., 2023; Hagendorff, 2023). For ToM evaluation specifically, the training corpora of LLMs may contain research papers detailing these psychological studies. Many past studies used identical or slightly altered language prompts to test LLMs, leading to potential contamination issues (Ullman, 2023). To critically evaluate the performance of LLMs on ToM tasks, researchers must have access to the datasets used to train them (Dodge et al., 2021), which are unfortunately not available.

**Shortcuts and spurious correlations.** The availability of shortcuts and spurious features has triggered many concerns that a model may leverage them to perform highly on a benchmark without robustly acquiring the desired skill (Sclar et al., 2023; Ullman, 2023; Shapira et al., 2023a). Recent findings suggest that LLMs tend to learn surface-level statistical correlations in compositional tasks, potentially leading to an illusion of systematic learning (Dziri et al., 2023). In all likelihood, LLMs are capable of learning ToM shortcuts in a similar manner.

## 3 Towards A Holistic Landscape of Machine Theory of Mind

### 3.1 Abilities in Theory of Mind Space (ATOMS) Framework

The evaluation of machine ToM lacks clarity and consistency across various literature, primarily due to the ambiguity surrounding the specific *mental states* being targeted. This ambiguity is not unique to the field of AI but is rooted in the complicated cognitive underpinnings of ToM. At the core of

this ambiguity is the latent nature of *mental states*, the subject has privileged access to them while others can only infer the existence of these mental states based on observable behaviors or expressions (Dretske, 1979; Blakemore and Decety, 2001; Zaki et al., 2009). Thus, it is impossible to directly access and assess the mental states of a human, and ToM must be tested indirectly through humans' ability to understand the relationship between mental states and behaviors, especially by predicting how agents behave based on their mental states (Swettenham, 1996; Phillips et al., 2002).

While the exact definition of ToM remains a central debate, the AI community can benefit from looking at what psychologists have viewed as an initial step. In this paper, we follow Beaudoin et al. (2020)'s taxonomy of ToM sub-domains, *i.e.,* the Abilities in Theory of Mind Space (ATOMS). As shown in Figure 1, the space consists of 7 categories of mental states, including *beliefs, intentions, desires, emotions, knowledge, percepts*, and *non-literal communication*. We selected this taxonomy because it was derived from a comprehensive meta-analysis of ToM studies. The meta-analysis focused on young children aged 0-5 years at the early stage of cognitive development, such that the setups are simpler and more comparable, avoiding complicated physical and social engagements that cannot be trivially deployed on LLMs.

**Beliefs.** Beliefs are informational states that people judge to be true, usually decoupled from motivational states (Dennett, 1995; Eccles and Wigfield, 2002). Beliefs, the most studied mental states in the field of ToM, are usually tested in the form of false belief tasks, including the unexpected contents test (Perner et al., 1987), the unexpected transfer (Sally-Anne) Test (Baron-Cohen et al., 1985), the second-order false belief (Icecream Van) Test (Perner and Wimmer, 1985). Researchers also studied their connection to actions and emotions (Swettenham, 1996).

**Intentions.** Intentions are choices with commitment, usually associated with concrete actions towards a goal (Cohen and Levesque, 1990). As a critical component of ToM, Kennington (2022) has called for a more explicit treatment of intentions. Intentions have been extensively explored in psychology tests, e.g., behavioral re-enactment (Meltzoff, 1995), action prediction (Phillips et al., 2002), intention explanation (Smiley, 2001), and intention attribution to abstract figures (Castelli, 2006).

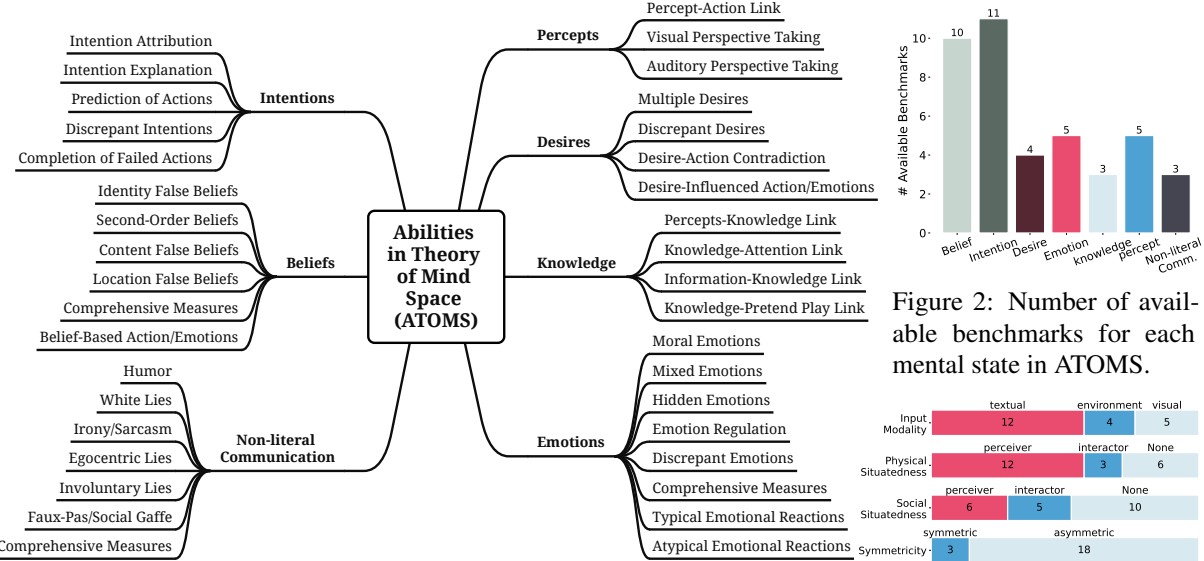

Figure 2: Number of available benchmarks for each mental state in ATOMS.

Figure 3: A comparison of benchmark settings.

Figure 1: The ATOMS framework of Beaudoin et al. (2020), which identified 7 categories of mental states through meta-analysis of ToM studies for children.

| Benchmarks and Task Formulations | Tested Agent | | | Situatedness | | | | ATOMS Mental States | | | | | | | | | Sym. |
|---|---|---|---|---|---|---|---|---|---|---|---|---|---|---|---|---|---|
| | Task | Input Modality | | Physical | | Social | | Belief | | Intention | | Des. | Emo. | Know. | Per. | NLC | |
| | | Text | Nonling. | Per. | Int. | Per. | Int. | 1st | 2nd+ | Act. | Com. | | | | | | |
| EPISTEMIC REASONING (Cohen, 2021) | Infer | T | - | | | | | ✓ | ✓ | | | | | | | | |
| TOMI (Nematzadeh et al., 2018) | QA | T | - | ✓ | | | | ✓ | ✓ | | | | | | | | |
| HI-TOM (He et al., 2023) | QA | T | - | ✓ | | | | ✓ | ✓ | | | | | | | | |
| MINDGAMES (Sileo and Lernould, 2023) | Infer | T | - | ✓ | | | | ✓ | ✓ | | | | | | ✓ | | |
| ADV-CSFB (Shapira et al., 2023a) | QA | H | - | ✓ | | | | ✓ | | | | | | | | | |
| CONVENTAIL (Zhang and Chai, 2010) | Infer | H | - | | | ✓ | | ✓ | | | ✓ | ✓ | | | | | |
| SOCIALIQA (Sap et al., 2019) | QA | H | - | | | ✓ | | | | ✓ | | | | ✓ | | | |
| BEST (Tracey et al., 2022) | - | H | - | | | ✓ | | ✓ | | | | | | ✓ | | | ✓ |
| FAUXPAS-EAI (Shapira et al., 2023b) | QA | H,AI | - | | | ✓ | | ✓ | | | | | | ✓ | | | ✓ |
| COKE (Wu et al., 2023) | NLG | AI | - | | | ✓ | ✓ | | | ✓ | | | | ✓ | | | |
| TOM-IN-AMC (Yu et al., 2022) | Infer | H | - | ✓ | | ✓ | | ✓ | ✓ | | | | | | | | |
| G4C (Zhou et al., 2023b) | NLG | H,AI | - | ✓ | | ✓ | ✓ | ✓ | ✓ | | | | | | ✓ | | |
| VISUALBELIEFS (Eysenbach et al., 2016) | Infer | - | Cartoon | ✓ | | | | ✓ | | | | | | | | | ✓ |
| TRIANGLE COPA (Gordon, 2016) | QA | H | Cartoon | | | ✓ | | | | ✓ | | | | ✓ | | | |
| MSED (Jia et al., 2022) | Infer | H | Images | ✓ | | | | | | | | ✓ | ✓ | | | | |
| BIB (Gandhi et al., 2021) | Infer | - | 2D Grid | ✓ | | | | | | ✓ | | ✓ | | | | | |
| AGENT (Shu et al., 2021) | Infer | - | 3D Sim. | ✓ | | | | | | ✓ | | ✓ | | | ✓ | | |
| MTOM (Rabinowitz et al., 2018) | Infer | - | 2D Grid | ✓ | | | | ✓ | | ✓ | | | | | | | |
| SYMMTOM (Sclar et al., 2022) | MARL | - | 2D Grid | ✓ | ✓ | ✓ | ✓ | | | | | | | ✓ | | | ✓ |
| MINDCRAFT (Bara et al., 2021) | Infer | H | 3D Sim. | ✓ | ✓ | ✓ | ✓ | | | ✓ | | | | ✓ | ✓ | | ✓ |
| CPA (Bara et al., 2023) | Infer | H | 3D Sim. | ✓ | ✓ | ✓ | ✓ | ✓ | ✓ | | | | | ✓ | ✓ | | ✓ |

Table 1: A taxonomized review of existing benchmarks for machine ToM and their settings under ATOMS. We further break **beliefs** into first-order beliefs (1st) and second-order beliefs or beyond (2nd+); and break **intentions** into Action intentions and Communicative intentions. **Tasks** are divided into Inference, Question Answering, Natural Language Generation, and MultiAgent Reinforcement Learning. **Input** modalities consist of Text (Human, AI, or Template) and Nonlinguistic ones. The latter further breaks into Cartoon, Natural Images, 2D Grid World, and 3D Simulation. The **Situatedness** is divided into None, Passive Perceiver, and Active Interactor. **Symmetricity** refers to whether the tested agent is co-situated and engaged in mutual interactions with other ToM agents.

**Desires.** Desires are motivational states that do not necessarily imply commitment, though they are usually emotionally charged and affect actions (Malle and Knobe, 2001; Kavanagh et al., 2005). Typical studies along this line include the Yummy-Yucky Task (Repacholi and Gopnik, 1997) for discrepant preferences from different individuals, the multiple desires within one individual (Bennett and Galpert, 1993), and the relationship between desires and emotions/actions (Wellman and Woolley, 1990; Colonnesi et al., 2008).

**Emotions.** Emotions are mental states associated with an individual's feelings and affective experiences, which could impact beliefs and behaviors (Frijda et al., 1986; Damasio, 2004). Most ToM studies on emotions focus on typical (Knafo et al., 2009) and atypical (Denham, 1986) emotional reactions to situations. Other studies also encompass affective perspective taking (Borke, 1971), understanding hidden emotions (Harris et al., 1986), and morally related emotions (Pons and Harris, 2000).

**Knowledge.** Many controversies revolve around the definition of knowledge as justified true beliefs (Gettier, 2000). In the context of AI, knowledge typically consists of information and organized representations of the world, which can be used to simplify understanding and address intricate reasoning and planning (Schank and Abelson, 2013). ToM studies usually involve understanding the absence of knowledge (Aronson and Golomb, 1999) as well as the connection between knowledge and perception (Ruffman and Olson, 1989) and attention (Moll et al., 2006).

**Percepts.** Humans are situated in the physical and social environments. To enable AI agents to operate in the world and communicate with humans, the sensory and social aspects of perception are crucial in a machine ToM. Along this line, psychological studies have investigated the perceptual perspective taking (Masangkay et al., 1974) and understanding the influence of limited perception on actions (Hadwin et al., 1997).

**Non-literal communications.** Being able to understand non-literal and figurative communication helps humans to perform pragmatic inference and reason about hidden words behind their written meanings (Giora, 2003). Non-literal communication has been recognized as an advanced ToM capability, spanning a wide spectrum of humor and deceptions (Happé, 1994), sarcasm (Sullivan et al., 1995), and faux-pas (social gaffe) situations (Baron-Cohen et al., 1999).

### 3.2 A Taxonomized Review of Benchmarks

The ATOMS framework can serve as an intuitive reference for researchers to identify their research priorities and situate their work better in the landscape of literature. We further take the initiative to provide a systematic review of existing benchmarks for machine ToM under the umbrella of ATOMS.[1] Although there are independent research initiatives on certain ToM facets like intention classification, emotion modeling, and aspects of non-literal communications, we primarily focus on those that explicitly target ToM or inferences of latent mental states. Besides the ToM dimensions in ATOMS, we further characterize the benchmarks on their task formulation, input modalities, physical and social situatedness, and symmetricity (whether the tested agent is co-situated and engaged in mutual interac-

tions with other ToM agents). We summarize our review in Table 1 and discuss our observations and under-explored aspects of ToM evaluation.

**Many aspects of ToM are under-explored.** As shown in Figure 2, we notice an overwhelming research focus on the intention and belief aspects of machine ToM. Several other aspects of ToM have not received enough attention. While the field of NLP has thoroughly explored different facets of emotion and non-literal communication, e.g., in the context of dialogue systems, ToM has rarely been explicitly mentioned as motivation. More connections and integrative efforts are clearly needed.

**Lack of clear targeted mental states.** Explicitly mentioning the Sally-Anne Test (Baron-Cohen et al., 1985) as inspiration, Grant et al. (2017) developed the predecessor of ToMI (Le et al., 2019). Similarly, Nematzadeh et al. (2018) cited the Icecream Van Test (Perner and Wimmer, 1985) as motivation and the FAUXPAS-EAI (Shapira et al., 2023b) benchmark followed the study of Baron-Cohen et al. (1999). While these benchmarks are cognitively grounded and target one particular aspect of ToM, the majority often incorporate multiple mental states without clear descriptions, which could make it challenging to measure the actual progress (Raji et al., 2021).

**Lack of situatedness in a physical and social environment.** Figure 6 illustrates the configurations of benchmarks. Each bar in the chart represents a distinct benchmark characteristic, and each segment within the bar illustrates the proportion of benchmarks with one specific setting. An immediate observation is a noticeable lack of benchmarks that encompass both physical and social environments, which highlights an existing research disparity in the field. We notice that many existing benchmarks are story-based, which verbalize the agent's perception of the environment and the behaviors of other agents in the form of story episodes, usually with language templates. The semantics of the environment are given by high-level events (e.g., Sally entered the kitchen). Many aspects of physical and social situatedness are overlooked in these benchmarks, e.g., spatial relations, the task and motivation of agents, and their action trajectories.

**Lack of engagement in environment.** We point out that existing benchmarks primarily adopt a passive observer role to test language agents. Yet the crucial aspects of interaction and engagement between the agent and other entities involved have

---

[1] We maintain a repository for relevant literature at `https://github.com/Mars-tin/awesome-theory-of-mind`.

been overlooked. Among all the benchmarks we reviewed, only three of them treat the tested model as an active agent, one that perceives the physical and social context, reasons about others' mental states, communicates with other agents, and interacts with the environment to complete pre-defined tasks (Sclar et al., 2022; Bara et al., 2021, 2023).

## 4 Towards A Situated Theory of Mind

### 4.1 Why A Situated ToM?

There have been concerns that cognitive inquiries are inadequate for gaining insight into understanding ToM for LLMs (Mitchell and Krakauer, 2023; Shapira et al., 2023a). However, we believe that the primary problem lies in using story-based probing as proxies for psychological tests, which situate human subjects in specific physical or social environments and record their responses to various cues. We, therefore, call for a situated evaluation of ToM, in which the tested LLMs are treated like agents who are physically situated in environments and socially situated in interactions with others.

**Situated evaluation covers more aspects of ToM.** Although it is possible to frame the situations as narratives and cover all mental states using text-only benchmarks, certain aspects of ToM can only be effectively studied within specific physical or social environment (Carruthers, 2015). This is because humans have the ability to infer the mental states of others through various modalities such as visual perception, actions, attention (gazes or gestures), and speech (Stack et al., 2022). For instance, studying perceptual disparities can be challenging with text-only datasets, as they often reduce complex scenarios to rule-based manipulations over negations in the prompts (Sileo and Lernould, 2023). Benchmarks that are not situated also face challenges when it comes to implementing coordination between agents, e.g., aligning intentions towards joint actions (Jain et al., 2019) and pragmatic generation (Zhu et al., 2021a; Bao et al., 2022).

**Situated evaluation mitigates data contamination.** A situated ToM evaluation can mitigate data contamination, as researchers can design scenarios in simulated settings that are unlikely to be part of the LLM's training data. Carefully designed benchmarks can also incorporate seen and unseen environments to assess generalization to new tasks and new environments, fundamentally addressing the issue of data contamination (Gandhi et al., 2021).

**Situated evaluation mitigates shortcuts.** By employing situated evaluation, the risk of taking shortcuts can be mitigated. Many of the existing ToM benchmarks are either limited in scale or adopt text templates to verbalize a (few) predefined scenario(s) and prompt LLMs for answers, giving answers away from syntactic structures and positional information (Le et al., 2019; Sclar et al., 2023). In a situated setting, on the contrary, we rely on simulated environments to manipulate evaluation data at scale, so that the environment, the states, and the action traces in the environment can be randomized to avoid the statistical spurious correlations. While situated evaluation can mitigate shortcuts, it does not eliminate the issue completely. For example, Aru et al. (2023) have reported that shortcuts can emerge in grid world setups if the design is not careful enough and randomness is limited. We emphasize that careful design and consideration are still required to curate any ToM benchmark.

### 4.2 A Preliminary Exploration in Grid World

In this section, we present a proof-of-concept study on a situated evaluation of ToM on LLMs. We choose to conduct our pilot study in Mini-Grid (Chevalier-Boisvert et al., 2018), a simple and commonly used environment for ToM studies in the machine learning community (Rabinowitz et al., 2018; Sclar et al., 2022). Through basic grid world representation, we can create tasks to challenge LLMs to reason about many aspects of physical and social situatedness, e.g., spatial relations, partial observability, agent's action trajectories, and from there, their beliefs, intent, emotions, etc. This is in stark contrast to existing story-based ToM benchmarks, which only contain high-level event episodes. We demonstrate that a diverse range of challenging ToM tests, covering all mental states from ATOMS, can be effectively created in a situated manner using a simple 2D grid world.

**Environment and Task Setups** We introduced 9 different ToM evaluation tasks for each mental state under ATOMS, and 1 reality-checking task to test LLMs' understanding of the world. It is important to acknowledge that our experiment serves as a proof of concept and does not aim to cover the entire spectrum of machine ToM, as our case studies are far from being exhaustive or systematic.

- **Reality Check**: Given the sequence of actions, predict the closest object at the end of the trajectory. The task is designed to test LLMs' under-

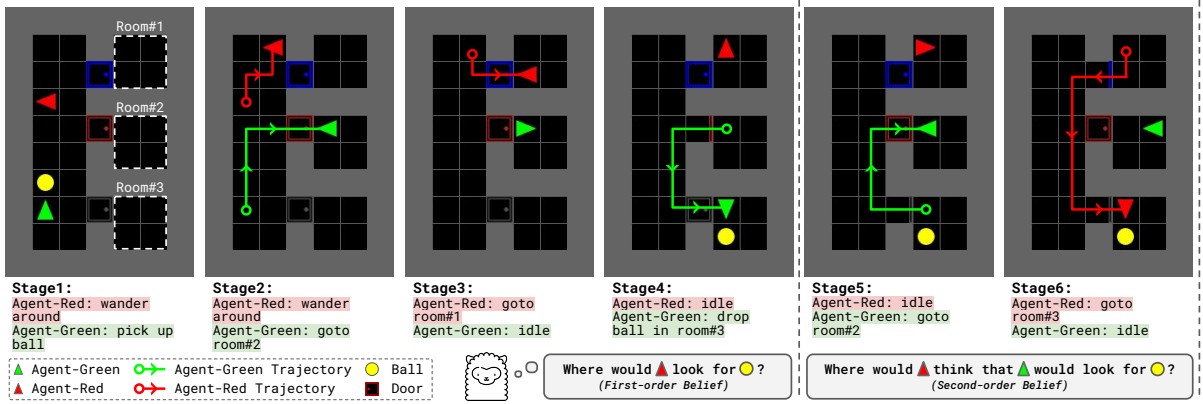

Figure 4: An overview of the first and second order false belief task illustrated in a grid world setup. We simulate the unexpected transfer scenarios with two agents, and verbalize the environment and action traces to test if LLMs hold a correct understanding of the agents' false beliefs.

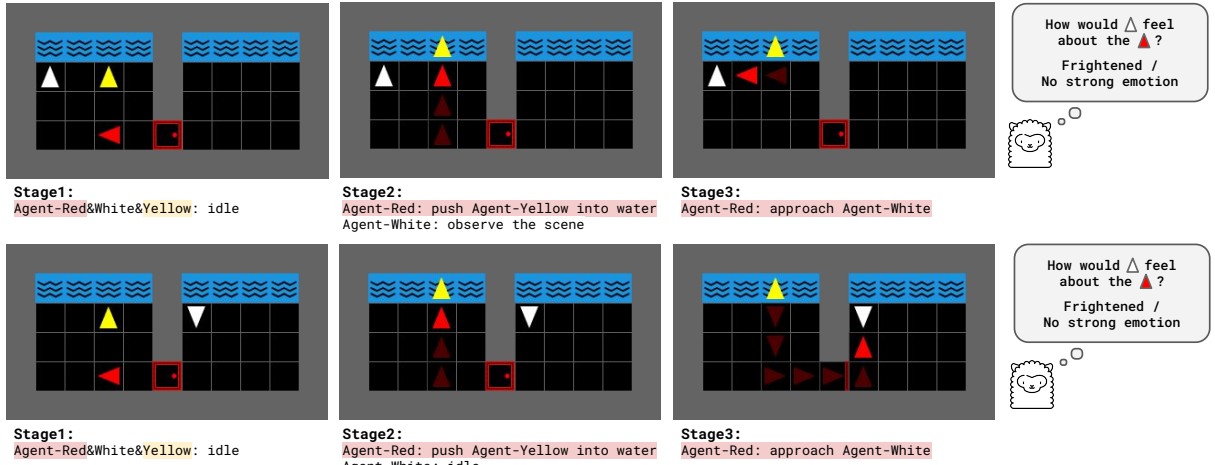

Figure 5: An overview of the morally related emotional reaction tasks illustrated in a grid world setup. We simulate scenarios where an agent either directly witnesses or is ignorant of a morally related event, and verbalize the environment and action traces to test if LLMs hold a correct prediction of the agent's emotional reaction.

standing of relocations in the grid world.

- **Short-term Intention**: Given an incomplete trajectory and a goal, predict the next action.
- **Long-term Intention**: Given an incomplete trajectory and a list of subgoals, predict the next subgoal that the agent is planning to achieve.
- **Desire**: Given a complete trajectory, predict if the agent demonstrates a preference for objects.
- **Percepts**: Given a complete trajectory, predict if the agent has a partial or full observation.
- **Belief**: The classic unexpected transfer task with possible first and second order false belief.
- **Non-literal Communication**: Given a trajectory and a statement from the agent, judge whether the agent is being deceptive.
- **Knowledge**: Given a trajectory, predict the object whose location is unknown to the agent.
- **Emotion**: The classic perception-emotion link test, where emotions are evoked in response to witnessing an emotionally stimulating situation.

We detail two case studies and leave examples of each task in Appendix A.

**Case Study 1: Beliefs.** Our belief experiments emulate the classic unexpected transfer tasks (Baron-Cohen et al., 1985; Perner and Wimmer, 1985). As is shown in Figure 4, we simulate this disparity of belief state and world state in MiniGrid. The first-order belief task features a main room with three connected side rooms, two agents named Red and Green, and a ball. Each instance of the belief experiment begins with Green placing the ball in Room#2 while Red watches. Red then enters a separate Room#1 and shuts the door. While Red is inside of this closed room, Green transfers the ball to Room#3. Red presumably holds a *false belief* about the location of the ball, believing it is in Room#2 though it is now in Room#3. Similarly, we implement the second-order belief task to test an incorrect belief that one agent holds about the belief of another. After Green has finished transferring the ball, it navigates to the room originally

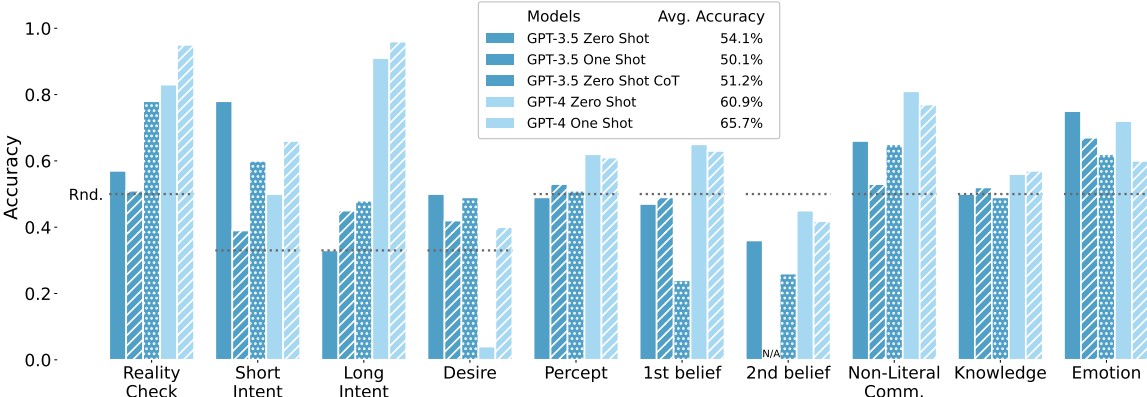

Figure 6: The LLMs' performance across the 10 tasks is illustrated. Each bar shows how one LLM performed with a specific prompting method. Overall, the tasks are tough for all the LLMs tested. While one-shot and CoT prompting can help in certain cases, their effectiveness is not consistent across the board.

containing the ball and shuts the door. Red then navigates to the room now containing the ball and sees the true location of the ball. Still, Green presumably possesses a false belief about Red's belief. In both tasks, LLMs are queried with two versions of the world: a false one with the ball in the original room, and a true one with the ball in the third room (its actual location). LLMs must correctly respond that the agents hold a false belief.

**Case Study 2: Emotions.** While the belief tasks highlight the importance of physical situatedness, we further demonstrate that social interactions can be simulated in the grid world. As is shown in Figure 5, We design morally related events that stimulate emotions (e.g., fear, appreciation). In this task, LLMs are queried to predict the emotional response of Agent-White, who either directly witnesses or is ignorant of this event. LLMs must correctly respond that the agent holds an emotional reaction only if it observes the event.

**Experiment Setups.** For each task, we create 100 instances following a prompt template that consists of [environment description], [agent description], [observability statement], [task statement], [actions sequences], [QA]. We select GPT-4 (gpt-4-0314) and Chat-GPT (gpt-3.5-turbo-0613) for evaluation on the 9 tasks.[2] Following prior work (Hu et al., 2022; Shapira et al., 2023a), we adopt MC-probing for LLMs that don't produce probabilities, which directly instructs LLMs to generate only the letter corresponding to the answer. Besides zero-shot evaluation, we also explored one-shot learning and Chain-of-Thought (CoT) prompting (Wei et al., 2022). More details are available in Appendix B.

---

[2]We use the ChatCompletion.create function from openai package.

**Results and Discussion.** We observe that LLMs exhibit some level of sensitivity for some mental states. Especially, GPT-4 scores up to 91% zero-shot accuracy and 96% one-shot accuracy in the long-term intention task. However, we also highlight the shortcomings of LLMs in some mental states of ATOMS to varying degrees, especially, in terms of predicting preferences, perception limitations, missing knowledge, and higher-order beliefs. These findings align with previous research (Sap et al., 2022; Trott et al., 2022; Shapira et al., 2023a), further confirming that LLMs are not yet reliable and comprehensive ToM agents. From the reality-checking task, we observe that GPT-3.5 reaches 78% accuracy with CoT prompting and GPT-4 significantly surpasses its predecessors with 83% zero-shot accuracy and 95% one-shot accuracy. Solving this reality check by no means implies that LLMs have a general perception ability of the real world, but that as a proof of concept, they demonstrate a certain (but still limited) level of situated awareness within the context of a basic abstract grid world. This implies that researchers can begin utilizing them as powerful building blocks for situated agents in complex ToM tasks. We note that it is always possible to come up with more challenging reality-checking questions to expose the limitations of LLMs, or to provide more guided prompts to assist LLMs in successfully completing ToM tasks. Undoubtedly, further research is required along this exciting yet challenging trajectory to advance ToM in LLMs and AI agents built upon LLMs.

## 5 Discussions and Action Items

### 5.1 The Scope of Machine Theory of Mind

**Be specific about the mental states studied.** Existing benchmarks often lack a clear target mental

state, making it challenging to interpret the results and measure the actual progress. To mitigate the risk of overestimating LLMs' ToM capabilities, it is recommended that future benchmark developers provide specific details regarding the targeted mental state(s) they intend to assess.

**Broaden the Scope of Machine ToM.** A breadth of mental states and their sub-domains have already been covered by AI benchmarks (Table 1). We observed an overwhelming emphasis on the benchmarks and modeling of *beliefs* and *intentions*, while other aspects have received insufficient attention. Still, there are considerably many blank spaces in the landscape of machine ToM, especially for more complicated forms of knowledge, desires, perspective-tasking, and emotional experiences beyond typical social situations.

## 5.2 Design New Theory of Mind Benchmarks

**Avoid shortcuts and spurious correlations.** The evaluation of LLMs itself presents significant challenges, not only in the case of ToM. Existing benchmarks suffer from issues such as data leakage and spurious correlations. Especially, shortcut solutions have been consistently reported in recent years (Le et al., 2019; Shapira et al., 2023a; Aru et al., 2023). We are in pressing need of new benchmarks with scalable sizes, high-quality human annotations, and privately held-out sets for evaluation.

**Avoid unfair evaluations from prompting.** Previous work has shown that CoT prompting can improve the performance of LLMs in ToM tasks (Li et al., 2023; Moghaddam and Honey, 2023; Shapira et al., 2023a). Various recent prompting mechanisms have also been developed to improve LLM's capability on ToM tasks (Zhou et al., 2023a; Leer et al., 2023). In the evaluation of LLMs' ToM capabilities, we recommend the careful documentation of prompts used and the avoidance of implicit human guidance to ensure a fair comparison.

**Move on to a situated ToM.** We call for a situated evaluation of ToM, in which the tested LLMs are treated like agents who are physically situated in environments and socially situated in interactions with others. A situated setup covers a wider range of ToM aspects. With carefully designed benchmarks with diverse environments and unseen test sets, a situated setup can help address data contamination issues and assess generalization to new tasks and environments. Furthermore, a situated setup allows for more complicated evaluation protocols than simple inference and QA tasks.

**Consider a mutual and symmetric ToM.** ToM is symmetric and mutual in nature, as it originally imputes the mental states of self and others. Prior research is largely limited to passive observer roles (Grant et al., 2017; Nematzadeh et al., 2018; Le et al., 2019; Rabinowitz et al., 2018) or speaker in a speaker-listener relationship (Zhu et al., 2021b; Zhou et al., 2023b). We encourage more studies on how humans and agents build and maintain common ground with a human ToM and a machine ToM through situated communication (Bara et al., 2021; Sclar et al., 2022). Besides, more research is needed to understand if LLMs possess early forms of intrinsic mental states given observation cues of the world. While we need to develop machines that impute the mental states of humans, humans should also develop a theory of AI's mind (ToAIM) (Chandrasekaran et al., 2017) by understanding the strengths, weaknesses, beliefs, and quirks of these black box language models.

## 5.3 Neural Language Acquisition and ToM

Both psychological studies (Bloom, 2002; Tomasello, 2005) and computational simulations (Liu et al., 2023) have demonstrated the effectiveness of ToM, especially intention, in language acquisition. Instead of concentrating on eliciting ToM in LLMs, we should contemplate whether certain ToM elements should be inherently present in LLMs or perhaps introduced alongside language pretraining. More research is needed to understand the connection between neural word acquisition and ToM development in machines.

## 6 Conclusion

In this position paper, we survey and summarize the ongoing debate regarding the presence of a machine ToM within LLMs, and identify the inadequate evaluation protocols as the roadblock. Many benchmarks focus only on a few aspects of ToM, and are prone to shortcuts. To mediate this issue, we follow the ATOMS framework to offer a holistic review of existing benchmarks and identify underexplored aspects of ToM. We further call for a situated evaluation of ToM, one that is physically situated in environments and socially situated in interactions with humans. We hope this work can facilitate future research towards LLMs as ToM agents, and offer an intuitive means for researchers to position their work in the landscape of ToM.

## Ethical Statement

The dataset created in this study includes instances that are synthetically generated from planners and RL algorithms, as well as ones created by humans. Human subjects research is approved by the University of Michigan Health Sciences and Behavioral Sciences Institutional Review Board (IRB-HSBS) under eResearch ID HUM00234647. The text generated by LLMs could potentially contain harmful, toxic, or offensive content. The authors have ensured that the data does not contain personally identifiable information or offensive content.

## Limitations

Our current benchmark only covers 100 instances for each task, adding up to only 1000 instances. Our experiment serves as a proof of concept and does not aim to cover the entire spectrum of machine ToM, as our case studies are far from being exhaustive or systematic. In the future, we plan to create a more systematic benchmark with a larger scale and various forms of evaluation. Additionally, it is worth noting that the ATOMS framework is derived from human ToM studies conducted with children under the age of 5. Consequently, this framework primarily focuses on the early developmental stages of ToM, capturing the naive and potentially rudimentary aspects of ToM. For more advanced ToM capability, we point to some recent frameworks proposed by Osterhaus and Bosacki (2022) and Stack et al. (2022).

## Acknowledgements

This work was supported in part by NSF IIS-1949634, NSF SES-2128623, and by the Automotive Research Center at the University of Michigan. Without implying any agreement with the contents as presented in this work, the authors extend their appreciation to Susan Gelman for her valuable feedback. The authors would like to thank all anonymous reviewers for their valuable feedback.

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

## A Task Settings and Data Collection

In this section, we provide an in-depth explanation of the ten tasks outlined in section 4.2. Task 0 serves as a "reality check" to assess LLMs' grasp of the physical world, particularly relocations within the grid world. Tasks 1 through 9 each emphasize distinct facets of ToM. All these tasks utilize MiniGrid, a streamlined 2D grid-world environment. (Chevalier-Boisvert et al., 2018).

### Task 0: Reality Check

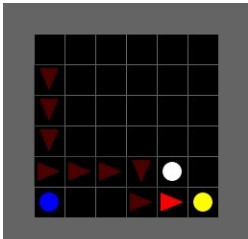

Figure 7: Illustration for *Reality Check* task

In this scenario, the agent is tasked with visiting 0-2 of the objects in the grid world. The agent has full observation in the world for efficient navigation. At least 2 objects (12 maximum) are placed in the environment. To test an LLM's ability to understand the physical actions taken by the agent, we ask it about the distance between the agent and various objects after it accomplishes a number of actions. The action planner is either a shortest path planner towards specified objects or a random action generator.

After the agent has finished 10 random actions, or right after it has visited one object with an optimal action planner, the task-related question will be generated with the following format:

**After having taken these actions, which item is the agent closer to?**

A. <object1.color> <object1.name>

B. <object2.color> <object2.name>

Here object1 and object2 both exist in the environment, and one of them is guaranteed to be the target object if there is such a goal. There are always two options for this task.

Along with the task-related question, the prompt includes a description of the grid world environment, the action space of the agent (only going forward, turn left / right in this task), a board-like depiction of the initial state, the list of actions taken

by the agent, and the agent's location and face direction for each step. For more details on prompting, please refer to section B.

The data for Task 0 are autonomously generated using seeds and a shortest-path planner.

### Task 1: Short Term Intention

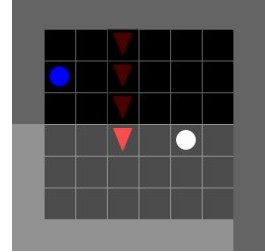

Figure 8: Illustration for *Short Term Intention* task

In this scenario, the agent is tasked with visiting either of the two objects in the grid world. The LLM is not provided with the goal object. Rather, it must determine the goal object by examining and understanding the agent's trajectory. In this task, the agent has full observation in the world, and there are exactly two objects placed. The object types and colors are randomly generated. The size of the room can be randomly sampled from the range 6 by 6 to 12 by 12.

To test an LLM's understanding of short-term intention, we ask it to predict the next action of the agent given its previous trajectory.

The agent's trajectory halts at a random step, with the exception of the precise moment when the optimal paths to the two objects diverge. This "cutting point" is set as an exception and also serves as the mean for the normal distribution from which the stopping point is sampled.

By restricting the cutting point in this manner, we guarantee the trajectory included in the prompt to be optimal for reaching the potential goal objects. This reduces the ambiguities of our experiments, and thus improves the significance of our results.

The task-related question has the following format:

**Which action will the agent take next?**

A. left

B. right

C. forward

The LLM should be able to choose which action would be next were the agent to continue its optimal trajectory to the goal object. The data for Task 1 are

autonomously generated using seeds and a shortest-path planner.

**Task 2: Long Term Intention**

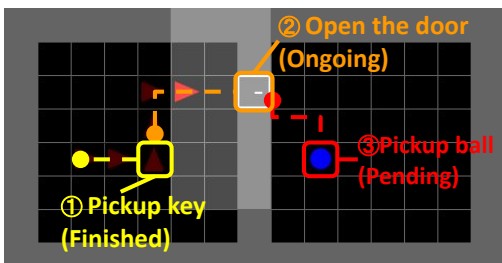

**Task setting:** complete three subgoals
in as few steps as possible
**Q:** which subgoal is the agent currently
trying to complete?
**A.** Locate and pick up a key
**B.** Locate and go through a door
**C.** Navigate to the object in the new
     room

Figure 9: Illustration for *Long term intention* task

In this scenario, the agent needs to complete the following subgoals in as few steps as possible: 1) Locate and pick up a key; 2) Locate and go through a door; 3) Navigate to the object in the new room. There are two rooms in this setting, which are connected by a locked door. The key of the door is always in the room in which the agent is initially located. The object is always placed in the other room.

We provide an LLM with a subset of the agent's trajectory and ask it which subgoal the agent is currently trying to complete.

The task-related question has the following format:

**Based on the agent's trajectory thus
far, which subgoal is the agent currently
trying to complete?**

A. Locate and pick up a key

B. Locate and go through a door

C. Navigate to the object in the new room

The data for Task 2 are autonomously generated using seeds and a shortest-path planner.

**Task 3: Desire**

In this scenario, the agent is required to pick up three objects as soon as possible. There are 2 types of objects in the world (e.g., blue balls and white balls), 3 of each. The agent may or may not have a preference for one object type (we stratify the data such that in 50% of the episodes the agent has

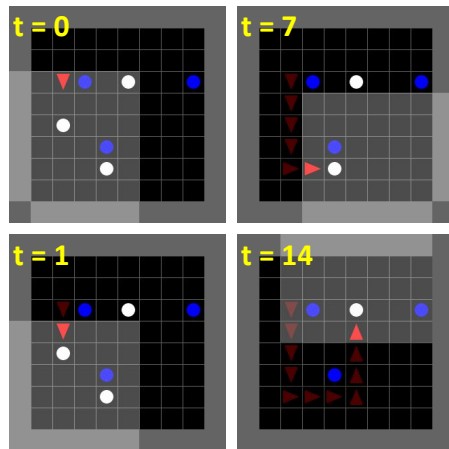

**Task setting:** pickup objects in the
environment. It may or may not have
its own preference over some objects.
**Q:** Which object (if any) does the
agent prefer?
**A.** Blue ball
**B.** White ball
**C.** No preference

Figure 10: Illustration for *Desire* task

a preferred object type and in the other 50% the agent lacks a preference). We also deduct from the final reward given to the agent when it takes a large number of steps to finish picking up three objects.

This task tests whether an LLM is able to determine the desire of the agent (for one object type or the other) by examining its trajectory. We prepared the following question for this task:

**Which object does the agent prefer?**

A. white ball

B. blue ball

C. no preference

The data for Task 3 are autonomously generated using seeds and Reinforcement Learning. We use the PPO algorithm (Schulman et al., 2017) to train the model. In the scenario wherein a preference is present, the preferred object type yields 10 times more reward than the non-preferred one. In the scenario wherein the preference is absent, both object types yield identical rewards.

**Task 4: Percept**

In this scenario, the agent is instructed to navigate in two rooms and reach the goal in the other room as fast as possible. In contrast with previous task settings, the agent has either a very limited visual range (3 x 3 grid in the front), or an "infinitely" large visual range (for practical purposes, the visual range is actually a 101 x 101 grid). Naturally, an agent with a smaller viewing range will

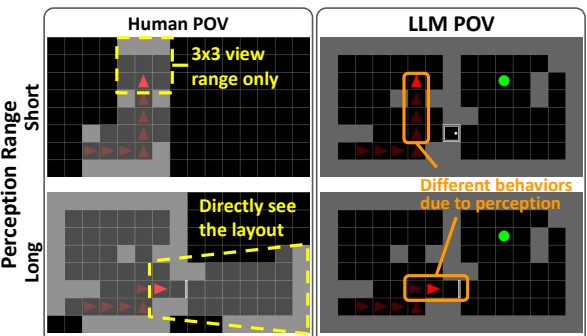

**Task setting:** navigate to the green ball with given perception range.
**Q:** Given only the environment layout and agent's trajectory, what is the most likely perceptual range that the agent possesses?
**A.** 3 blocks
**B.** Infinitely many blocks
*The perceptual range refers to the number of cells that an agent can see in front of it

Figure 11: Illustration for *Percept* task

make more mistakes (e.g., navigating to a dead end) while trying to reach the goal object. Obstacles are randomly placed in each room to block the agent's view.

LLMs are expected to only look at the trajectory of the agent in an environment, and determine whether the agent has a limited view range or a nearly full view range. The question format is as follows:

**Based on the agent's actions, what is the most likely perceptual range that the agent possesses? The perceptual range refers to the number of cells that an agent can see in front of it.**

A. 3 blocks

B. infinitely many blocks

We manually collected 100 trials in total for both situations.

## Task 5 & 6: First and Second Order Belief

In this scenario, there are two agents in the environment. Both of them are initially in the main room (on the left side of the whole grid world; see Figure 4). On the right side, there are three small rooms. Agents can freely go in and out of each room. They can see everything inside the current room and can see the other rooms through the door if it is open.

This task reproduces the unexpected transfer (Sally-Anne) test (Baron-Cohen et al., 1985; Perner and Wimmer, 1985), with both first-order and second-order belief checking. In the first-order belief task, one agent does not see the second agent

transfer a ball from one room to another. Presumably, therefore, the agent falsely believes the ball to be in its previous location. The second-order belief task extends the first-order belief task by enabling the agent with the false belief to see the ball in its new location. The other agent, however, does not witness the first agent rectifying its false belief, so it presumably holds a second-order false belief (about the belief state of the first agent). By varying the observations that each agent makes (e.g., whether or not each agent sees the transfer of the ball from one room to the another) as well as the order of these events, this task setting allows us to check an LLM's first-order and second-order belief capabilities.

Rather than asking the LLM directly where to find the object, we provide two board-like belief states as options. We then query the LLM about which board-like state the agent is more likely to believe.

Data for tasks 5 & 6 are collected via rule-based planners with several scenarios.

## Task 7: Non-Literal Communication

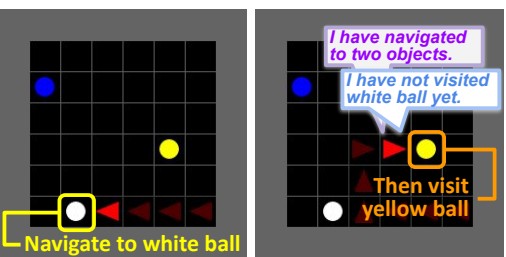

**Task setting:** visiting all the objects, and report its progress with texts.
**Q:** The agent claims that it at one point navigated to blue ball. Based on the agent's actions, is it telling the truth?
**A.** Yes
**B.** No

Figure 12: Illustration for *Non-literal communication* task

This task focuses specifically on one form of non-literal communication: lying. Within this task, the LLM is told explicitly that there is an agent tasked with navigating to all of the objects within the grid world. In each instance of the task, however, the agent only visits a subset of the objects in the environment. The LLM is subsequently told that the agent has claimed success in visiting a particular object. This object is randomly selected so that sometimes it is an object that has actually been visited and other times it is not.

The question format is as follows:

**Based on the agent's actions, is it telling the truth?**

A. yes

B. no

To successfully complete this task, the LLM must combine its knowledge about the physical occurrences taking place within the grid world with its knowledge about lying, a vital component in Beaudoin et al. (2020)'s category of non-literal communication.

**Task 8: Knowledge**

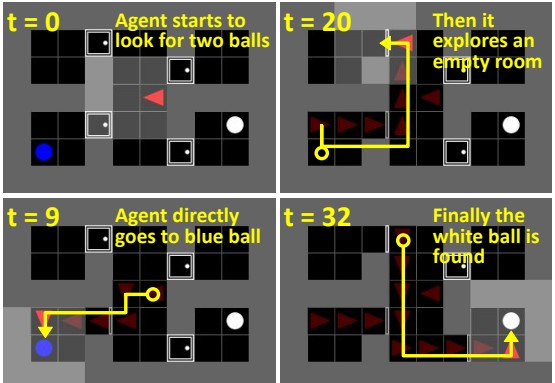

```
Task setting: pick up both two objects in the
environment with limited visibility. One of their
location is known. The other is unknown. LLM will
only see the environment layout and agent's
trajectory.
Q: Based on the agent's actions, does it know the
position of blue ball before?
A.  Yes
B.  No
```

Figure 13: Illustration for *Knowledge* task

This task requires the agent to pick up two objects that exist in the environment. Both two objects are known to be placed separately in two of the four rooms.

In this scenario, the agent is informed of the location of one object, while the location of the second object remains unknown. The agent is instructed to collect the objects in a specific sequence. Ideally, if the agent knows an object's location, it should proceed directly to the appropriate room. Otherwise, it will have to search for the object in the yet-to-be-explored rooms.

We include the agent's entire trajectory in the prompt. We then ask the LLM to determine if the agent knows the position of one object before. The question format is presented as follows:

**Based on the agent's actions, does**

**it know the position of <color> <name> before?**

A. Yes

B. No

The data for Task 8 are autonomously generated using seeds and a rule-based planner.

**Task 9: Emotions**

This task (see Figure 5) requires LLMs to infer the emotions of agents in a situated context from their physical behaviors. Specifically, it involves variations on a theme involving a small lake of water. In every variation, one agent pushes another agent into the lake. In some of these variations, an observer is privy to the situation. The prompt then asks the LLM about this observer's feelings towards both the victim and the perpetrator. Presumably, the observer should experience sympathy for the victim and anger (or a similarly negative emotion) for the perpetrator.

**How would <observer.name> most likely feel about <pusher.name>?**

A. no strong emotion

B. angry

In other variations, a helper comes along and pulls the victim out of the lake. Presumably, the observer should feel positive emotions (e.g. respect, gratitude) for this helper. The question format is as follows:

**How would <observer.name> most likely feel about <helper.name>?**

A. no strong emotion

B. respectful

## B   Prompting and Reproducibility

In this pilot study, our data curation follows a uniform structure across all tasks similar to prior work (Li et al., 2022), deviating only slightly to account for task-specific circumstances.[3]

**Environment Description**   Each prompt begins with a description of the two-dimensional world wherein the task will take place. Specifically, our prompting code provides LLMs with the dimensions of the game board and a method to reference specific cells (column-first Cartesian coordinates).

---

[3]The data for this pilot study is available at https://huggingface.co/datasets/sled-umich/2D-ATOMS.

Each prompt subsequently itemizes the various objects that are situated in the world, along with their coordinates and attributes. Although this prompting structure could be easily adapted to handle many different types of attributes, we focused only on color for the sake of simplicity. Additionally, this section assigns each object a "label": a single letter that the prompt uses to represent the object in a printed grid representation.

**Agent Description, Observability, and Task**
The next section of each prompt is a detailed description of the agent(s) occupying the grid world. In the example below, which was taken from task 1, only one agent occupies the grid world. In multi-agent tasks, this section details the various properties of all agents. Whether single or multi-agent, however, the basic properties are the same. The prompt first details the position and direction of the agent. It is located in a specific cell, and it always faces up, down, left, or right. The prompt then specifies the various actions that an agent is capable of taking (e.g. "forward", "left", and "open"). The agent's labels are then specified. Within the printed grid representation, the agent is always represented as a V-like shape depending on the direction that it is facing. For instance, the prompting tool uses < to represent an agent that is facing left. Finally, the prompt specifies two key attributes of the agent: (1) its level of observability (e.g. whether or not it can see into adjacent rooms) and (2) its goal. Sometimes these two descriptors are heavily modified, restricted, or removed altogether so that they do not interfere with the task. For instance, when testing LLMs for percepts, the prompt does not specify the visual range of the agent.

**Action Sequences** Following the agent description section, the prompt prints out a board representation, a multi-line sequence of plain text that appears two-dimensional when printed out. Here, the various objects and agents are depicted in position by their associated labels. Additionally, the configuration of the walls is specified by a perimeter of W's. Next, the prompt specifies a sequence of actions that take place over the course of the episode being considered by the LLM. In the episode depicted below, the agent navigates part of the way to a red box. These actions always specify the new position and orientation of the agent.

**Questions and Answer Candidates** Finally, each prompt contains a question that the LLM must answer. These questions are the most task-specific portion of the prompt, so their contents vary, however, they are always multiple-choice. Additionally, they always contain a set of instructions below heeding the LLM to return only a single letter in its response.

This is a grid-like 2D world
The grid world consists of 6 rows and
6 columns, 0-based
We use (i,j) to represent the i-th
column (from left to right) and j-th
row (from top to bottom).

The following is a list of objects in
this world. Each line starts with the
object's position and is followed by
its attributes
(2, 3): key, grey; represented by this
label: G
(4, 4): box, red; represented by this
label: H

Walls are depicted using the symbol W

There is an agent at (2, 2) facing
left

The agent can take the following
actions:
- left: makes the agent face left of
where it is currently facing
- right: makes the agent face right
of where it is currently facing
- forward: makes the agent move one
step in the direction it is currently
facing
- open: makes the agent open a door
that it is in front of
- pickup: makes the agent pick up
the object that it is in front of
- drop: makes the agent drop an item
that it is holding
- stay: makes the agent stay where it
currently is for a timestep

The agent is represented by the
following labels depending on which
direction it is facing:
- Facing left: <
- Facing up: ^
- Facing right: >
- Facing down: v

The agent has full observability,
meaning it can see the entire world

The agent has been instructed to
navigate to one of the two objects
in the environment, although you do
not know which

This is the starting state of the
board:
```
    0 1 2 3 4 5
0 | W W W W W W
1 | W O O O O W
2 | W O < O O W
3 | W O G O O W
4 | W O O O H W
5 | W W W W W W
```
This list contains a sequence of
actions taken by the agent
(Step 1) The agent took action left
and is now at (2, 2) facing down
(Step 2) The agent took action left
and is now at (2, 2) facing right
(Step 3) The agent took action forward
and is now at (3, 2) facing right
(Step 4) The agent took action forward
and is now at (4, 2) facing right
(Step 5) The agent took action right
and is now at (4, 2) facing down

Which action will the agent take next?
A: left
B: right
C: forward

Please ONLY respond using the letter
corresponding to your answer
Do not generate any text other
than the letter