# OpenReview forum: "Towards A Holistic Landscape of Situated Theory of Mind in Large Language Models"
_EMNLP/2023/Conference — EMNLP 2023 Findings_

### Official Review · Reviewer_92YE · 2023-08-03

**Soundness:** 4

**Excitement:**

4: Strong: This paper deepens the understanding of some phenomenon or lowers the barriers to an existing research direction.

**Paper Topic And Main Contributions:**

Large Language Models (LLMs) have generated considerable interest and debate regarding the potential emergence of a machine Theory of Mind (ToM). This paper seek to answer two road-blocking questions: (1) How can we taxonomize a holistic landscape of machine ToM? (2) What is a more effective evaluation protocol for machine ToM to avoid spurious correlations? They follow the ATOMS framework to offer a holistic review of existing benchmarks and identify under-explored aspects of ToM. They further call for a situated evaluation of ToM, one that is physically situated in environments and socially situated in interactions with humans. They conduct a preliminary study in the grid world scenario is presented over 10 different task setups as a proof of concept.This paper provides a deep understanding in the landscape of ToM and effective observations for future research.

**Reasons To Accept:**

The content of this paper is solid and comprehensive, which is useful for NLP community. I suggest this paper should be presented at the conference.

**Reasons To Reject:**

I didn't see any obvious weakness.

**Reproducibility:**

3: Could reproduce the results with some difficulty. The settings of parameters are underspecified or subjectively determined; the training/evaluation data are not widely available.

**Reviewer Confidence:**

3: Pretty sure, but there's a chance I missed something. Although I have a good feel for this area in general, I did not carefully check the paper's details, e.g., the math, experimental design, or novelty.

---

> ### Author Rebuttal · Authors · 2023-08-29
>
> We thank the reviewer for finding our paper “solid and comprehensive” and “useful for NLP community”, and we thank the reviewer for the positive feedback that:
> > This paper provides a deep understanding in the landscape of ToM and effective observations for future research.
>
> We will make our data and prompts available for the community to ensure the reproducibility of our pilot study.

---

### Official Review · Reviewer_HssE · 2023-08-04

**Soundness:** 3

**Excitement:**

4: Strong: This paper deepens the understanding of some phenomenon or lowers the barriers to an existing research direction.

**Paper Topic And Main Contributions:**

With the advent of large language models, there has been an ongoing debate on whether Machine Theory of Mind (ToM) has emerged in LLMs. This position paper calls for a holistic and situated evaluation of ToM in LLMs.

The paper provides a survey and summary of existing works, presenting both evidence and counter-evidence for the emergence of a machine ToM in LLMs. The paper proposes to taxonomize ToM into different mental states and evaluate it based on these states. Specifically, the paper follows ATOMS to taxonomize machine ToM into 7 mental states and performs a holistic review of existing ToM benchmarks to identify under-explored aspects of ToM. Moreover, the paper calls for a situated evaluation of ToM, one that is physically situated in environments and socially situated in interactions with humans. As part of its contribution, the paper also presents a preliminary study in the text grid world as a proof of concept.

In conclusion, this paper offers a comprehensive analysis of existing literature on ToM in LLMs and emphasizes the need for a situated evaluation benchmark and a taxonomy of ToM into different mental states for evaluation.

**Questions For The Authors:**

A: It is unclear to me how the conclusion “LLMs have reached a level of maturity where researchers can begin utilizing them as powerful building blocks for situated agents in complex ToM tasks.” (L524-527) is derived.
Evaluating ToM in situated physical environments poses a challenge related to the model's ability to perceive the physical environment accurately. If an LLM performs poorly at perceiving the physical environment, it becomes difficult to evaluate its ToM capability. In the preliminary study, the paper proposes the task “Reality Check” for testing LLM’s understanding of the world. This task involves predicting the closest object around the end of the trajectory. However, it is hard to conclude that good performance on this task reflects the model's ability to understand the world. Therefore, I find it hard to derive the stated conclusion about the maturity of LLMs for situated agents in complex ToM tasks.

B: Furthermore, Figure 5 shows that only GPT4 outperforms the random baseline for the "Reality Check" task, while ChatGPT also achieves random performance on the "Reality Check" task. Therefore it is difficult for me to see why “LLMs have reached a level of maturity where researchers can begin utilizing them as powerful building blocks for situated agents in complex ToM tasks.” Also, to provide a more comprehensive analysis, the paper could investigate the performance of other open-sourced LLMs.

**Reasons To Accept:**

- The paper performs a comprehensive review of existing works on machine ToM in LLMs, presenting both evidence and counter-evidence for the emergence of ToM.
- The paper proposes to taxonomize ToM into different mental states and evaluate it based on these mental states in order to ensure clarity and consistency in evaluating ToM across different literature.
- The paper performs a taxonomized review of existing ToM benchmarks and identifies under-explored aspects of existing ToM benchmarks.
- The paper motivates the need for a situated evaluation of ToM, one that is physically situated in environments and socially situated in interactions with humans.
- The paper presents a preliminary study that showcases a practical demonstration of constructing evaluation tasks related to 7 different aspects of ToM under the setting of a text-based grid world.

**Reasons To Reject:**

- The paper lacks an in-depth discussion on how a situated evaluation could mitigate shortcuts and spurious correlations. The section of L438-448 (Situated evaluation mitigates shortcuts) appears to focus more on how situated evaluation can help with data contamination (L193) through the configuration of a vast number of diverse environments and complex tasks. However, it does not provide a detailed analysis of how a situated evaluation can help mitigate the issue of shortcuts and spurious correlations.
- The paper presents an interesting preliminary study as a proof-of-concept study on a situated evaluation of ToM in LLMs. Since, to me, it is questionable if the text-based grid world setting truly resembles the physically situated and socially situated environment, and therefore it is unclear to me what conclusion or insights the paper seeks to draw from this study.

**Reproducibility:**

4: Could mostly reproduce the results, but there may be some variation because of sample variance or minor variations in their interpretation of the protocol or method.

**Reviewer Confidence:**

3: Pretty sure, but there's a chance I missed something. Although I have a good feel for this area in general, I did not carefully check the paper's details, e.g., the math, experimental design, or novelty.

**Typos Grammar Style And Presentation Improvements:**

Typos:
- L380: “Figure 5” -> “Figure 3”

Presentation Improvements:
- Unclear Corresponding Characteristics: In L381 - 384, the paper mentions that “Each layer within the figure corresponds to a specific characteristic of the benchmark”. However, in the same paragraph, the paper does not provide a clear description of what the corresponding characteristics are. The reader is left to infer from L350-351, which suggests the characteristics could be related to “task formulation, input modalities, physical and social situatedness, and symmetricity”. To improve clarity, the paper should explicitly describe the corresponding characteristics for each layer in Figure 3. Also, in the case where the inner layer's characteristic is "input modalities," it is unclear why "env-only" is considered an orthogonal setting to "text-only" and "multimodal."
-  The specific meaning of the colors used in Figure 3 is not clear from the paper. The paper should improve Figure 3 and provide clear descriptions.

---

> ### Author Rebuttal · Authors · 2023-08-29
>
> We thank the reviewer for recognizing many aspects of our contributions and consider our work as “comprehensive”. We appreciate the in-depth review, and would like to make the following responses to clarify some misunderstandings. We are looking forward to the reviewer’s further comments during the discussion period.
>
> ### Responses to Weaknesses
>
> > **W1**: The paper lacks an in-depth discussion on how a situated evaluation could mitigate shortcuts and spurious correlations. The section of L438-448 (Situated evaluation mitigates shortcuts) appears to focus more on how situated evaluation can help with data contamination (L193) through the configuration of a vast number of diverse environments and complex tasks. However, it does not provide a detailed analysis of how a situated evaluation can help mitigate the issue of shortcuts and spurious correlations.
>
> We thank the reviewer for raising this concern about the clarity of our discussion!
>
> We agree that L438-448 is more about data contamination than shortcuts. A situated ToM evaluation can mitigate data contamination as the grid world (and possibly more complex environment) are not likely to be included in the training data of LLM.  A shortcut introduced by training/testing splits in previous works on benchmarks does not apply here, as we do not pre-train any LLMs.
>
> The shortcuts are taken at the evaluation time. Many of the existing ToM benchmarks adopt text templates to verbalize a (few) predefined scenario(s) and prompt LLMs for response. Researchers have shown that LLMs can get the correct answer by utilizing syntactic structures or positional information, especially in few-shot settings. In our situated ToM setting, we can use simulated environment to manipulate evaluation data, so that the environment, the states, the trajectories in the environment can be randomized at scale, and we can avoid the statistical spurious correlations.
>
> We emphasize that while our proposed situated ToM evaluation can mitigate shortcuts, it does not eliminate the issue completely. First, we echo Aru et al. (2023) that shortcuts can still emerge in grid world setups if the design is not careful enough and randomness is limited (line 440). We note that careful consideration is required to design any benchmark, which is not specific to ToM. Also, we mentioned that shortcuts can be introduced in the few-shot examples and human prompts introduced at inference time (line 563 and below).
>
> We will add more detailed discussion in the revised version.
>
> > **W2**: The paper presents an interesting preliminary study as a proof-of-concept study on a situated evaluation of ToM in LLMs. Since, to me, it is questionable if the text-based grid world setting truly resembles the physically situated and socially situated environment, and therefore it is unclear to me what conclusion or insights the paper seeks to draw from this study.
>
> We thank the reviewer for raising this concern about the motivation for using text-based grid world for proof of concept! There are two reasons we specifically chose text-based grid environment for our investigations.
>
> First, text-based grid world provides a good abstraction of the physical world in a simplified environment to study many aspects of embodied agents. It has been used by several recent works, e.g., for grounding [1], planning and decision making [2], interaction and language acquisition [3], etc.
>
> Second, situated tasks involve perception and reasoning about the surroundings (e.g., through computer vision). In the grid world, the low-level world state (e.g., the grid setup and where the agent is in the grid) can be captured by symbols. This setup allows us to focus on ToM reasoning without being distracted by machine perception. Through basic grid world representation, the agent is tasked to reason many aspects of **physical situatedness**, e.g., spatial relations between agents, their states, their action trajectories, and from there, the goals, intent, emotions, etc. This is in stark contrast from the existing story-based ToM where the semantics of the world is given by natural language stories without requiring the agent to reason about the world (e.g., based on the change of the world and traces of action trajectories).
>
> [1] A Benchmark for Systematic Generalization in Grounded Language Understanding. In NeurIPS 2020. https://arxiv.org/abs/2003.05161
>
> [2] Pre-Trained Language Models for Interactive Decision-Making. In NeurIPS 2022. https://arxiv.org/abs/2202.01771
>
> [3] BabyAI: A Platform to Study the Sample Efficiency of Grounded Language Learning. In ICLR 2019. https://arxiv.org/abs/1810.08272
>
> ### Responses to Questions
>
> > **Q1**: It is unclear to me how the conclusion “LLMs have reached a level of maturity where researchers can begin utilizing them as powerful building blocks for situated agents in complex ToM tasks.” (L524-527) is derived. Evaluating ToM in situated physical environments poses a challenge related to the model's ability to perceive the physical environment accurately. If an LLM performs poorly at perceiving the physical environment, it becomes difficult to evaluate its ToM capability. In the preliminary study, the paper proposes the task “Reality Check” for testing LLM’s understanding of the world. This task involves predicting the closest object around the end of the trajectory. However, it is hard to conclude that good performance on this task reflects the model's ability to understand the world. Therefore, I find it hard to derive the stated conclusion about the maturity of LLMs for situated agents in complex ToM tasks.
>
> Sorry for the confusion caused by this statement. We will rephrase it in the next iteration. We mainly want to say that LLMs have demonstrated incredible performance in a range of NLP tasks. Given an increasing amount of work in applying LLMs in ToM, it’s time to systematically study LLMs for situated ToM.
>
> - **We are not trying to say they already reach maturity for ToM**, but rather they are at the stage where we can put situated ToM evaluation on the table. We totally agree with the reviewer regarding perception which plays an integral role in reasoning, which is also one of the motivations for calling for “situated ToM”.  The frame of “situatedness” is really trying to bring perception into the picture of ToM. Also, as the reviewer pointed out, poor perception will make it difficult to evcaluate ToM. This is exactly one of the reasons we chose the grid-world setup as a proof of concept here. Future work will need to take more complex forms of perception into account.
>
> - **We are not trying to say they already reach maturity for world understanding**. Reality check here is a simplified setting which serves as a reference pointing with respect to other tasks.  Solving this reality check by no means implies the agent’s ability in general perception of the real world. With our current experimental setup, we show that LLMs like GPT-4 can to some extent solve some simplified perception tasks (Reality Check) but can also fail some ToM tasks in similar protocol. We will add the clarifications in the revised version.
>
> > **Q2.1**: Furthermore, Figure 5 shows that only GPT4 outperforms the random baseline for the "Reality Check" task, while ChatGPT also achieves random performance on the "Reality Check" task. Therefore it is difficult for me to see why “LLMs have reached a level of maturity where researchers can begin utilizing them as powerful building blocks for situated agents in complex ToM tasks.”
>
> We apologize for the confusion on the performance of ChatGPT in reality checking. Here we provide additional results that ChatGPT can obtain 78% performance on reality check using zero-shot Chain-of-Thoughts (CoT) reasoning, compared to 57% under zero-shot multiple choice (MC) probing. While CoT significantly improves ChatGPT’s ability in reality checking, many ToM tasks don’t benefit (and sometimes suffer) from CoT reasoning due to false rationale.
>
> | **Case** | **Reality Check** | **Short Intent** | **Long Intent** | **Desire** | **Percept** | **1st belief** | **2nd belief** | **NL Com.** | **Knowledge** | **Emotion** |
> | ----------- | ----------- | ------------ | ---------- | ---------- | ---------- | ---------- | ---------- | ---------- | ---------- | ---------- |
> | GPT3.5 zero shot **MC** | 57% | 78% | 33% | 50% | 49% | 47% | 24% | 66% | 62% | 75% |
> | GPT3.5 zero shot **CoT** | 78% | 60% | 48% | 49% | 51% | 24% | 16% | 65% | 52% | 62% |
>
> > **Q2.2**: Also, to provide a more comprehensive analysis, the paper could investigate the performance of other open-sourced LLMs.
>
> As this is a position paper rather than a benchmark/analysis paper, the goal of our preliminary experiment is to present a proof of concept and showcase possibilities to design situated ToM benchmarks using 2D grid world and evaluate LLMs on it. We believe that more work is indeed needed in the future to design more comprehensive ToM benchmarks and test more variants of LLMs.
>
> Still, we provide some more results on public LLMs upon the reviewer’s request. We conducted further study to compare the results based on LlaMA-7B (no human alignment) and Vicuna-7B (with human alignment) on 30 randomly selected test for each task (due to the time constrain of the rebuttal period) as shown below. We use LM-probing to select the choice with the highest probability following Brown et al. 2020 and Shapira et al. 2023a. Surprisingly, we found that these models make similar predictions on the same task, leading to similar performance. We see that human alignment may have a limited impact on situated ToM capabilities for relatively smaller LLMs (i.e., 7B models).
>
> | **LLM** | **Reality Check** | **Short Intent** | **Long Intent** | **Desire** | **Percept** | **1st belief** | **2nd belief** | **NL Com.** | **Knowledge** | **Emotion** |
> | ----------- | ----------- | ------------ | ---------- | ---------- | ---------- | ---------- | ---------- | ---------- | ---------- | ---------- |
> |  LLaMA  | 46.67% | 16.67% | 26.67% | 50.00% | 43.33% | 50.00% | 46.47% | 33.33% | 40.00% | 44.44% |
> | Vicuna | 46.67% | 16.67% | 26.67% | 50.00% | 43.33% | 50.00% | 50.00% | 33.33% | 40.00% | 44.44% |
>
> ### Presentation Improvements
>
> We thank the reviewer for the helpful suggestions regarding presentation improvements. We will add detailed descriptions using the additional page and redo Figure 3 to make it clearer.

---

### Official Review · Reviewer_myQj · 2023-08-05

**Soundness:** 3

**Excitement:**

3: Ambivalent: It has merits (e.g., it reports state-of-the-art results, the idea is nice), but there are key weaknesses (e.g., it describes incremental work), and it can significantly benefit from another round of revision. However, I won't object to accepting it if my co-reviewers champion it.

**Missing References:**

Park et al., 2023. Generative Agents: Interactive Simulacra of Human Behavior

**Paper Topic And Main Contributions:**

This paper surveys and studies theory of mind in large language models, by first introducing previous evidence in finding and not finding theory of minds in LLMs, and then point out the limitations in previous evaluation. The authors then follow the Abilities in Theory of Minds Space (ATOMS) to validate the current ToM benchmarks, and propose a situated evaluation of ToM in the grid world scenario consisting of 10 different tasks. The tasks include beliefs, intentions, desires, emotions, knowledge, perception, and non-literal communications defined in ATOMS, and showed that similar to previous findings, certain but limited level of world understanding presents in LLMs.

**Questions For The Authors:**

1. For data contamination, why would it be necessary to have access to the datasets used to train LLMs before evaluating the performance of LLMs on ToM tasks? There are tons of benchmarks used for LLM evaluation, why would ToM tasks be different from others?
2. In line 257-261, why would ATOMS be developed from young children, but "cannot be easily deployed on state-of-the-art AI agents"?
3. Do you also input the graphs in the 10 tasks as illustrated in Figure 4 to GPT-4/ChatGPT in your experiment setup? If not (according to the appendix in my understanding), how are the images used?
4. Why do you choose to use a zero-shot setup for evaluation instead of few-shot prompting?


**Reasons To Accept:**

1. As many researchers are trying to find connections between powerful LLMs and theory of mind, this paper provides a structured review of existing task formulation and evaluation, and pointed out the current limitations from a clear taxonomy perspective.
2. More importantly, the paper advocates a situated evaluation of ToM by first proposing a prototype environment. This can be interesting to researchers studying theory of mind in the community.

**Reasons To Reject:**

1. Although the community may benefit from the study and advocated holistic landscape by resolving some of the current limitations, it is not convincing from the paper why each part of the taxonomy derived from 0-5 years old children, would be necessary for evaluating theory of mind in a large language model. More importantly, it is not clear why the proposed 10 tasks would be better at evaluating theory of mind compared to all other baselines (e.g., emotion), especially when there is no detailed results and analysis in either the main paper or the appendix.
2.  Theory of mind evaluation may be impacted by RLHF in large language models, but the situated ToM evaluation (at least from the 10 tasks introduced in the paper), are not related to how the LLMs would respond to different aspects defined.

**Reproducibility:**

2: Would be hard pressed to reproduce the results. The contribution depends on data that are simply not available outside the author's institution or consortium; not enough details are provided.

**Reviewer Confidence:**

4: Quite sure. I tried to check the important points carefully. It's unlikely, though conceivable, that I missed something that should affect my ratings.

---

> ### Author Rebuttal · Authors · 2023-08-29
>
> We thank the reviewer for recognizing our contributions (“structured review for existing formulation, evaluation, and limitations”, “situated evaluation that can be interesting to researchers studying theory of mind in the community”). We appreciate their constructive feedback on our efforts, and would like to make the following responses. We are looking forward to the reviewer’s further comments during the discussion period.
>
> ### Responses to Weaknesses
>
> > **W1.1**: Although the community may benefit from the study and advocated holistic landscape by resolving some of the current limitations, it is not convincing from the paper why each part of the taxonomy derived from 0-5 years old children would be necessary for evaluating theory of mind in a large language model.
>
> The taxonomy is based on developmental psychological studies, as these aspects have been considered as a part of human’s ToM and are demonstrated during human’s early development (e.g., before 5 years old). ToM evaluations are often conducted on pre-school children to detect early signs of developmental problems (e.g., autism).  While possessing these 7 mental states does not imply a complete human-level ToM, a lack of understanding for any of these mental states can imply a limitation in LLM’s ToM capability.
>
> > **W1.2**: More importantly, it is not clear why the proposed 10 tasks would be better at evaluating theory of mind compared to all other baselines (e.g., emotion), especially when there is no detailed results and analysis in either the main paper or the appendix.
>
> We believe a situated perspective can cover more aspects of ToM and avoid a crucial portion of shortcuts introduced by textual templates and data contamination (section 4.1). The goal of this paper is to offer a new perspective of evaluating ToM in large models and showcase examples of designing such benchmarks. The proposed 10 tasks are used as a proof of concept to demonstrate how to systematically and holistically study situated ToM that covers many aspects of mental states, which go beyond individual types of mental states (e.g., emotion as raised by the reviewer) which are separately studied in the past.
>
> Position papers often focus on general discussions on the state-of-the-art, key limitations, and future directions. Here we go beyond the traditional form and use an example as a proof of concept to supplement our discussion. We will certainly add more details about this example in the revision.
>
> > **W2**: Theory of mind evaluation may be impacted by RLHF in large language models, but the situated ToM evaluation (at least from the 10 tasks introduced in the paper), are not related to how the LLMs would respond to different aspects defined.
> We are not sure what the review meant by “situated ToM evaluation are not related to how the LLMs would respond”. We try our best to clarify with the following aspects, and would appreciate any additional clarifications from the reviewer.
>
> - **ToM evaluation and RLHF**: While previous research shows that RLHF improves LLM’s understanding of user’s instruction intention, which is an aspect of LLM’s ToM **capability**, we are not aware how existing ToM **evaluation** can be affected by RLHF. To the best of our knowledge, we compare the LLMs before and after RLHF on their performance on ToM tasks, which also applies to our proposed situated ToM evaluation methods.
> - **Does human alignment improve LLM’s situated ToM?**: Previous research shows that RLHF improves LLM’s understanding of user’s instruction intention, which is only one aspect of LLM’s ToM **capability**. We use 10 situated tasks to model a variety of ToM which go beyond instruction intention. We conducted further study to evaluate the role of human alignment on these tasks. To do so, we compare the results based on LlaMA-7B (no alignment) and Vicuna-7B (with alignment) on 30 randomly selected test for each task (due to the time constrain of the rebuttal period) as shown below. We use LM-probing to select the choice with the highest probability following Brown et al. 2020 and Shapira et al. 2023a. Surprisingly, we found that these models make similar predictions on the same task, leading to similar performance. We see that human alignment may have a limited impact on situated ToM capabilities for relatively smaller LLMs (i.e., 7B models).
>
> | **LLM** | **Reality Check** | **Short Intent** | **Long Intent** | **Desire** | **Percept** | **1st belief** | **2nd belief** | **NL Com.** | **Knowledge** | **Emotion** |
> | ----------- | ----------- | ------------ | ---------- | ---------- | ---------- | ---------- | ---------- | ---------- | ---------- | ---------- |
> |  LLaMA-7B  | 46.67% | 16.67% | 26.67% | 50.00% | 43.33% | 50.00% | 46.47% | 33.33% | 40.00% | 44.44% |
> | Vicuna-7B | 46.67% | 16.67% | 26.67% | 50.00% | 43.33% | 50.00% | 50.00% | 33.33% | 40.00% | 44.44% |
>
> As we are not sure if we completely understand the reviewer’s comment, we would appreciate additional clarification if our answers do not address the concern.
>
> ### Responses to Questions
>
> > **Q1**: For data contamination, why would it be necessary to have access to the datasets used to train LLMs before evaluating the performance of LLMs on ToM tasks? There are tons of benchmarks used for LLM evaluation, why would ToM tasks be different from others?
>
> Data contamination is one of the major concerns for evaluating pre-trained models. There is no exception for LLM as they are trained in internet scale data, and we don’t know if the model has access to the test set during pretraining. Indeed it’s the case that there are tons of benchmark tasks (including existing story-based ToM benchmarks), due to the nature of the closed models, we have to take these evaluations with a grain of salt (Anna Roger’s blog attested to that: https://hackingsemantics.xyz/2023/closed-baselines/).
>
> More specifically to ToM evaluation, the training corpora of LLMs contain many research articles describing cognitive tests and psychological studies. Many of the previous studies employed the same language prompt for testing children, or only made minimal change of wording. This leads to a data leakage issue in LLM’s ToM evaluation. This concern was shared by many previous works, e.g., Ullman 2023.
>
> Our situated ToM Task, using simple grid-world setup, intends to alleviate the potential problem of data contamination.
>
>
> > **Q2**: In line 257-261, why would ATOMS be developed from young children, but "cannot be easily deployed on state-of-the-art AI agents"?
> We believe that there might be a misunderstanding of our original statement, we mean exactly the opposite:
> > The meta-analysis has focused on young children aged 0-5 years at the early stage of their cognitive development, and the experimental setups could be relatively simple and comparable, **without** complicated social interactions that cannot be easily deployed on state-of-the-art AI agents.
>
> An extensive body of previous works has shown that very basic skills, e.g., commonsense reasoning, both physical commonsense and social commonsense, which are developed at a very young age, turned out to be a notorious problem for AI agents (e.g, many works from Yejin Choi’s group have demonstrated that). We will provide more discussion to clarify this statement.
>
> As ATOMS was derived from early children research, human studies are usually in simple forms of visual and verbal prompts, which can be adapted to LLMs. In more advanced ToM studies, complex social interactions and decision making are involved, which cannot be transferred into LLM evaluation without additional computational modeling.
>
>
> > **Q3**: Do you also input the graphs in the 10 tasks as illustrated in Figure 4 to GPT-4/ChatGPT in your experiment setup? If not (according to the appendix in my understanding), how are the images used?
> We did not use the image of the grid world, but a symbolic representation of the grid world in the form of text. This is a common practice to prompt LLMs for similar 2D grid world from prior work, e.g., [1].
>
> [1] Pre-Trained Language Models for Interactive Decision-Making. In NeurIPS 2022. https://arxiv.org/abs/2202.01771
>
> > **Q4**: Why do you choose to use a zero-shot setup for evaluation instead of few-shot prompting?
>
>  The zero-shot setup is the best to consider as a first step [2] for evaluating LLM’s ability. As this is a position paper rather than a benchmark paper, we only include zero shot as a proof of concept. We believe that more work is needed in the future to design more comprehensive sets of ToM benchmarks and test more methods to apply LLMs.
>
> Still, we present the one-shot results upon the reviewer’s request. We found that many ToM tasks benefit from one shot of demonstration, while some suffer from this setting.
>
> | **Case** | **Reality Check** | **Short Intent** | **Long Intent** | **Desire** | **Percept** | **1st belief** | **2nd belief** | **NL Com.** | **Knowledge** | **Emotion** |
> | ----------- | ----------- | ------------ | ---------- | ---------- | ---------- | ---------- | ---------- | ---------- | ---------- | ---------- |
> | **GPT3.5 zero shot**  | 57% | 78% | 33% | 50% | 49% | 47% | 24% | 66% | 62% | 75% |
> | **GPT3.5 one shot**  | 51% | 39% | 45% | 42% | 53% | 49% | N/A* | 53% | 47% | 67% |
> | **GPT4 zero shot**  | 83% | 50% | 91% | 4% | 62% | 65% | 45% | 81% | 59% | 72% |
> | **GPT4 one shot**  | 95% | 66% | 96% | 40% | 61% | 63% | 45% | 77% | 53% | 60% |
>
> *N/A as the prompt exceeded the maximal context window.
>
> [2] Best practices for prompt engineering with OpenAI API. By OpenAI, https://help.openai.com/en/articles/6654000-best-practices-for-prompt-engineering-with-openai-api

---

### Meta-Review · Area_Chair_spUx · 2023-09-19

**Recommendation:** 3

**Metareview:**

Overall, this paper addresses the growing discussion of ToM in the context of LMs, working towards a more robust evaluation protocol.
The reviewers overall slantly positive on the paper, but don't necessarily arrive at the strongest consensus on its strengths and weaknesses.

Collectively, as AC I see the following points as the clearest strengths and weaknesses:
S: Rigor/structured approach to ToM, concrete and situated evaluations
W: Unclear whether the approach to eval (derived from children + 10 tasks initiated based on taxonomy) make sense, concerns about spurious correlations are not fully addressed

In a brief reading of the paper, I agree that the paper makes important progress on a timely topic with substantive soundness. I will say I do share the concern that matter of spurious correlations, which seems to be at the heart of why evaluations for ToM in particular will be challenging to design + trust, has not been adequately resolved. If accepted, I encourage the authors to spend the majority of the additional page strengthening this topic. Separately, I think the community would be collectively more excited/inspired by the work if the results offered more striking analysis: I am left a bit unwhelemd by the results/discussion of results. I understand a well-designed evaluation is itself a great service to the community, but I think making the results (and/or their analysis) richer can really help draw interest to the evaluation as well, which is likely of interest for driving future work in this space.

---

### Decision · Program_Chairs · 2023-10-07

**Decision:**

Accept-Findings

**Comment:**

Overall, this paper addresses the growing discussion of ToM in the context of LMs, working towards a more robust evaluation protocol.
The reviewers overall slantly positive on the paper, but don't necessarily arrive at the strongest consensus on its strengths and weaknesses.

Collectively, as AC I see the following points as the clearest strengths and weaknesses:
S: Rigor/structured approach to ToM, concrete and situated evaluations
W: Unclear whether the approach to eval (derived from children + 10 tasks initiated based on taxonomy) make sense, concerns about spurious correlations are not fully addressed

In a brief reading of the paper, I agree that the paper makes important progress on a timely topic with substantive soundness. I will say I do share the concern that matter of spurious correlations, which seems to be at the heart of why evaluations for ToM in particular will be challenging to design + trust, has not been adequately resolved. If accepted, I encourage the authors to spend the majority of the additional page strengthening this topic. Separately, I think the community would be collectively more excited/inspired by the work if the results offered more striking analysis: I am left a bit unwhelemd by the results/discussion of results. I understand a well-designed evaluation is itself a great service to the community, but I think making the results (and/or their analysis) richer can really help draw interest to the evaluation as well, which is likely of interest for driving future work in this space.